# A Low-Complexity Hand Gesture Recognition Framework via Dual mmWave FMCW Radar System

**DOI:** 10.3390/s23208551

**Published:** 2023-10-18

**Authors:** Yinzhe Mao, Lou Zhao, Chunshan Liu, Minhao Ling

**Affiliations:** School of Communication Engineering, Hangzhou Dianzi University, Hangzhou 310018, China; mao212080186@hdu.edu.cn (Y.M.); chunshan.liu@hdu.edu.cn (C.L.); 212080155@hdu.edu.cn (M.L.)

**Keywords:** gesture recognition, mmWave FMCW radar, dual-radar system, signal processing, deep learning

## Abstract

In this paper, we propose a novel low-complexity hand gesture recognition framework via a multiple Frequency Modulation Continuous Wave (FMCW) radar-based sensing system. In this considered system, two radars are deployed distributively to acquire motion vectors from different observation perspectives. We first independently extract reflection points of the interested target from different radars by applying the proposed neighboring reflection points detection method after processing the traditional 2-dimensional Fast Fourier Transform (2D-FFT). The obtained sufficient corresponding information of detected reflection points, e.g., distances, velocities, and angle information, can be exploited to synthesize motion velocity vectors to achieve a high signal-to-noise ratio (SNR) performance, which does not require knowledge of the relative position of the two radars. Furthermore, we utilize a long short-term memory (LSTM) network as well as the synthesized motion velocity vectors to classify different gestures, which can achieve a significantly high accuracy of gesture recognition with a 1600-sample data set, e.g., 98.0%. The experimental results also illustrate the robustness of the proposed gesture recognition systems, e.g., changing the environment background and adding new gesture performers.

## 1. Introduction

Hand gestures are one of the most effective communication means for human beings and have been used in various applications for human–computer interaction (HCI) [1], including smart homes [2], virtual reality (VR) [3], game control [4], and text input [5]. In these applications, accurate gesture recognition plays an important role in enabling sophisticated functionalities and ensuring user experience. As such, many existing works have attempted to develop gesture recognition solutions, for which there are mainly three categories, i.e., wearable device-based solutions [6], visual device-based solutions [7], and wireless sensing-based solutions [8].

Wearable device-based solutions rely on the sensors equipped on wearable devices to collect motion data in real time [9,10,11]. However, the requirement of wearing the devices can lead to annoying inconveniences and hence may limit their applications. Visual device-based solutions rely on camera sensors for gesture recognition [12,13,14] and can be applied in a diverse range of scenarios [15,16,17] due to their exceptional experiential qualities. The main drawback of a visual device-based solution is its potential possibility of the invasion of privacy.

Wireless sensing-based solutions deal with the above problems by measuring and analyzing radio signals that interact with human gestures [8]. One popular sensing-based solution relies on WiFi systems [5,18,19,20], which can classify human gestures via Doppler features extracted from the channel state information (CSI) obtained at WiFi transceivers [21]. One potential limiting factor of WiFi-based systems is that obtaining CSIs from commercial WiFi chips may be difficult since most commercial WiFi CSI tools are proprietary of chip manufactures [22] and are not open.

As an alternative sensing-based gesture recognition solution, mmWave radar-based systems [23,24,25,26] have recently received significant attention due to the recent advances in mmWave radar chip manufacturing [27] and the continuing reduction in the cost and physical size of hardware [28]. Unlike WiFi-based solutions, mmWave radar-based solutions do not need to stick to the CSI measurement intervals specified by WiFi standard [22], hence allowing for more flexible measurements for hand gesture recognition. Moreover, as mmWave radar operates in a dedicated frequency band that is different from most existing communication systems, the impact of mutual interference is less severe.

This works focuses on mmWave radar-based systems for hand gesture recognition. Specifically, we consider mmWave Frequency Modulation Continuous Wave (FMCW) radar for hand gesture recognition, motivated by its capability of simultaneously measuring the range and velocity of the target of interest. Unlike most FMCW-based systems that consider the use of a single FMCW-radar to extract the features of a hand gesture, in this work, we consider a dual FMCW system that utilizes two FMCW radars to measure the signals of human hand gestures. We note that as the chip size and cost are continuing to drop for mmWave FMCW radars, it is possible to deploy multiple mmWave FMCW radar sensors in the same space, e.g., in a smart home, to support diverse applications such as human activity detection, health monitoring, and hand gesture recognition.

To exploit the benefit of two mmWave FMCW radars, we first develop an algorithm to extract the motion velocity vectors of human hand gestures based on the fusion of range and velocity information obtained from the multiple radars. As will be shown later in this paper, the motion velocity vector acts as an distinct feature that can be used to classify different human gestures. Based on the motion velocity vector extracted, a gesture classification model based on a long short-term memory (LSTM) network is proposed to capture the sequential dependence of the velocities. Our main contributions are highlighted below:We develop a systematic framework of high-accuracy hand gesture recognition based on two mmWave FMCW radars. The proposed framework first detects the moving target and identifies the reflection points of the target from each frame of range-Doppler (r-D) images generated from the two radars. The angular information of the target is then obtained by applying beamforming. Then, the motion velocity vector is synthesized by combining the angles and velocities of the target estimated by the two radars.We propose a novel feature extraction algorithm to synthesize the motion velocity vector that is crucial for hand gesture recognition. The proposed algorithm can effectively estimate the true velocity of the moving target and eliminates the requirement of the knowledge on the relative position of the two radars. This allows for flexible deployment of the two radarsWe propose a deep-learning gesture classification model based on the LSTM network, which effectively captures the sequential correlations between the motion velocities at different time instances. The proposed model takes only the motion velocity vector as the input feature and is shown to achieve satisfactory hand gesture recognition accuracy and robustness. In an experiment contributed by 10 volunteers with 1600 samples, the proposed model achieves 98.0% accuracy.

The rest of the article is organized as follows. Section 2 discusses related work with a focus on existing FMCW-based hand gesture recognition solutions. Section 3 presents some basic principles of FMCW radar processing Section 3. Our proposed hand gesture solution is detailed in Section 4. In Section 5, we present experiment results and provide a comparison with traditional approaches. Conclusions are drawn in Section 6. For ease of exposition, vectors are denoted by bold lowercase characters, whereas matrices are represented by bold uppercase characters.

## 2. Related Work

Classification algorithms for hand gesture recognition are mainly divided into two categories: machine learning-based solutions and deep learning-based solutions. At the early stage of FMCW-based hand gesture recognition, machine learning is the mainstream method for classification. The method in [23], proposed by the Google team, extracts nine low-dimensional motion features and applies a Random Forest classifier to identify gestures for HCI applications. In addition, Malysa et al. [24] proposed gathering motion feature vectors using a 77 GHz FMCW radar and training them based on Hidden Markov Models (HMM).

With the development of neural networks, there are numerous hand gesture recognition systems incorporating neural networks into their designed hand gesture recognition systems, and the Convolutional Neural Network (CNN) is one of the most widely adopted algorithms. As a representative example, ref. [29] proposed generating micro-Doppler (m-D) images from the radar signals and then using a pre-trained CNN model for hand gesture classification. This approach is a straightforward application of the procedure commonly adopted in imaging processing, but for hand gesture classification, its classification accuracy is limited since the m-D images cannot distinguish between different gestures with similar image patterns, e.g., the gesture of swiping from left to right and the gesture of swiping from right to left. To resolve this issue,  ref. [30] proposed a multi-stream CNN model that takes three types of figures as the input, i.e., the range-time figure, the Doppler-time figure, and the angle-time figure, and achieves better classification accuracy. The authors of [31] further improved the accuracy of gesture recognition by exploiting the range-Doppler-angular multi-dimensional gesture features space and exploring optimal multi-dimensional feature combination strategies for a multi-channel CNN. In addition, based on their previous work, the authors in [32] proposed the method of spatial position alignment and adopted a multi-channel CNN for multi-position hand gesture recognition.

The LSTM network is extensively employed as another prominent method in hand recognition systems. For example, the authors of [33] extracted motion profiles from r-D images and used the LSTM encoder to complete the final recognition by extracting the global temporal features. Meanwhile, the authors of [34] obtained the range-Doppler-angle trajectories from a 77 GHz FMCW Multiple Input Multiple Output (MIMO) radar, and built a LSTM network with the reused forward propagation approach to learn the gesture features.

## 3. Preliminaries

### 3.1. Radar Signal Model

In this subsection, we briefly introduce the principle of the FMCW radar. The transmitted signal of the FMCW radar consists of a series of continuous frequency-modulated waves. The signal within a frequency modulation period is called a chirp. The transmitted FMCW signal frame consists of multiple chirps; see Figure 1 for an illustration, and all notations are listed in Table 1.

For a single chirp, the transmitted signal can be expressed as [31]: (1)$$\begin{array}{c} \begin{array}{c} s_{T}\left(t\right)=A_{T}\exp\left[j2\pi \left(f_{0}t+\int_{0}^{t}K\tau d\tau \right)\right], \end{array} \end{array}$$
where *t* is the fast time, $t\in [0,T_{\mathrm{chirp}}]$, $A_{T}$ denotes the complex amplitude coefficient of the transmitted signal, and *K* is the frequency modulation slope.

The signal echoed from a moving target and received by the radar can be represented as [31]:  
(2)$$\begin{array}{c} \begin{array}{c} s_{R}\left(t\right)=A_{R}\exp\left\{j2\pi [\left(f_{0}+f_{d}\right)\left(t-\Delta t\right)+\int_{\Delta t}^{t}\left[K\left(\tau -\Delta t\right)\right]d\tau ]\right\}, \end{array} \end{array}$$
where $A_{R}$ is the complex amplitude coefficient of the received signal, $f_{d}=\frac{2v}{\lambda }=\frac{2vf_{c}}{c}$ denotes the Doppler shift, λ is the wavelength, $f_{c}$ is the carrier frequency of the transmitted signal, *v* is the velocity, $\Delta t=\frac{2d}{c}$ is the flight time of the signal, *c* is the speed of light, and *d* denotes the range of the target.

The received signal $s_{R}\left(t\right)$ is mixed with the transmitted signal upon receiving it to produce the intermediate frequency (IF) signal, which can be represented as [31]: (3)$$\begin{array}{c} \begin{array}{c} y\left(t\right)=s_{T}\times s_{R}^{*}=A_{T}A_{R}\cdot \exp\left\{j2\pi \left(f_{0}+f_{d}\right)\Delta t+\left(K\Delta t-f_{d}\right)t-K\Delta t^{2}/2\right]\}. \end{array} \end{array}$$

Since $K\Delta t^{2}/2$ is typically much smaller than other terms in Equation (Equation 3), the IF signal can be approximated as: (4)$$\begin{array}{c} \begin{array}{c} y\left(t\right)\approx A\cdot \exp\left\{j\left[2\pi \left(f-f_{d}\right)t+\varphi \right]\right\}, \end{array} \end{array}$$
where $A=A_{T}\cdot A_{R}$, f=KΔt is the frequency of the IF signal and
(5)$$\begin{array}{c} \begin{array}{cc} \varphi & =2\pi \left(f_{c}+f_{d}\right)\cdot \Delta t=\left(f_{c}+f_{d}\right)\cdot \frac{4\pi d}{c}\approx \frac{4\pi d}{\lambda }, \end{array} \end{array}$$
is the time-varying phase.

For different chirps in the same frame, *f* and φ will change from one chirp to another. This can be used to estimate of the velocity and the distance of the target of interest. Supposing the IF signal is sampled at a fixed time interval $T_{s}$, then the discrete-time samples of a frame can be stored in a two-dimensional matrix $Y\in C^{N\times L}$, for which each column corresponds to a different chirp. Here, *N* is the number of samples in each chirp and *L* is the number of chirps in a frame. Following Equation (Equation 4), the entries of Y can be represented as: (6)$$\begin{array}{c} \begin{array}{c} Y\left[n,l\right]=A\cdot \exp\{j\left[2\pi \left(f_{l}-f_{d}\right)nT_{s}+\varphi_{l}\right]\}, \end{array} \end{array}$$
where *n* is the fast time index, standing for the index of sampling points in the modulation time, n=1,2,…N, and *l* is the slow time index, corresponding to the chirp index in a frame, l=1,2,…,L. $f_{l}$ and $\varphi_{l}$ are related to the range of target $d_{l}$ in the *l*-th chirp.

### 3.2. Range and Velocity Estimation

To extract the range information of the target, Fast Fourier Transform (FFT) is applied to each column of Y. For the *l*-th chirp (column), the frequency domain signal can be represented as: (7)$$\begin{array}{c} \begin{array}{c} {\hat{y}}_{l}\left[\omega \right]=A\exp\left(j\varphi_{l}\right)\delta \left[\omega -\left(f_{l}-f_{d}\right)T_{s}\right]\approx A\exp\left(j\varphi_{l}\right)\delta \left(\omega -f_{l}T_{s}\right). \end{array} \end{array}$$

After the FFT operation, the range of the target can be estimated according to the peak index of the frequency response, denoted as $\eta_{l}$, and the relationship between the $f_{l}$, $T_{\mathrm{chirp}}$, and $\eta_{l}$ can be expressed as follows: (8)$$\begin{array}{c} \begin{array}{c} no\eta_{l}=f_{l}/\frac{1}{T_{s}\cdot N}=f_{l}\cdot T_{\mathrm{chirp}}, \end{array} \end{array}$$
where $f_{l}=K\Delta t=2Kd_{l}/c$. Then, the range between the radar and the target can be calculated as:  
(9)$$\begin{array}{c} \begin{array}{c} d_{l}=\frac{\eta_{l}\cdot c}{2KT_{c}}=\eta_{l}\cdot \frac{c}{2B}. \end{array} \end{array}$$

To estimate the velocity of the target, FFT is applied to each row of matrix Y. For the *n*-th row, the IF signal in Equation (Equation 6) can be rewritten as: (10)$$\begin{array}{c} \begin{array}{cc} y_{n}\left[l\right]= & A\cdot \exp\left\{j\left[2\pi \left(f_{l}-f_{d}\right)nT_{s}\right]\right\}\cdot \exp\left\{j\left[\varphi^{\prime }+\left(l-1\right)\Delta \varphi \right]\right\}, \end{array} \end{array}$$
where $\varphi^{\prime }$ is the phase of $y_{n}\left[1\right]$, Δφ is the phase difference of two adjacent elements of $y_{n}$, and $\Delta \varphi =4\pi vT_{\mathrm{chirp}}/\lambda$. For the *n*-th row, the difference between adjacent values of *f* is much smaller than Δφ; it can be approximated as $f_{1}$. Then, the signal after the FFT operation can be represented as: (11)$$\begin{array}{c} \begin{array}{cc} {\hat{y}}_{n}\left[\xi \right]=A & \cdot \exp\left\{j\left[2\pi \left(f_{1}-f_{d}\right)nT_{s}\right]+\varphi^{\prime }\right\}\cdot \delta \left[\xi -\frac{2vT_{\mathrm{chirp}}}{\lambda }\right]. \end{array} \end{array}$$

Denoting $\mu_{n}$ as the frequency index of the peak in the spectrum, the velocity of the target can be calculated as: (12)$$\begin{array}{c} \begin{array}{c} v_{n}=\frac{\mu_{n}\lambda }{2T_{\mathrm{chirp}}}. \end{array} \end{array}$$

### 3.3. Direction of Target Movement

For hand gesture recognition, it is intuitive that the direction of the hand movement is an important feature. However, such a feature is not directly available from single FMCW radar signals. To see this, let us consider a simple example, as illustrated in Figure 2, where a reflecting target *P* moves in the field of view (FoV) of the radar with velocity *v* towards direction α. We assume that the FMCW radar has multiple antennas. Thus, the instantaneous direction of the target with respect to the radar, i.e., θ, can be estimated via array signal processing techniques [35].

The velocity of the motion can be decomposed into the radial velocity, $v_{n}$, and the tangent velocity, $v_{\tau }$. The radial velocity, $v_{n}$, can be expressed as [18]: (13)$$\begin{array}{c} \begin{array}{c} v_{n}=\sin\left(\pi -\left(\alpha +\theta \right)\right)\cdot v=\sin\left(\alpha +\theta \right)\cdot v. \end{array} \end{array}$$

As demonstrated, the azimuth angle, θ, and the direction of motion, α, are the primary factors that influence $v_{n}$.

The velocity estimated by the principles mentioned in Section 3.2 correspond to the radial velocity $v_{n}$ since it is estimated based on the Doppler principle. The tangent velocity $v_{\tau }$, which may play an important role in gesture recognition, cannot be estimated. Motivated by this issue, we propose to utilize two mmWave FMCW radars with a MIMO antenna array.

## 4. The Dual FMCW Radar System for Gesture Recognition

In this section, we introduce the dual FMCW radar system for gesture recognition and detail the signal processing procedure. The dual-radar system consists of two mmWave FMCW radars that are placed at the same height, with their FoVs partially overlapped. The two radars are both connected to the same controller (e.g., a computer) for FMCW control and synchronization. The IF signals sampled from these two radars are both collected at the controller for further signal processing.

Figure 3 illustrates the overall procedure of the hand gesture recognition system. The IF signals are first processed to remove the static part of the clutter by eliminating mean value of phase, followed by 2-dimensional Fast Fourier Transform (2D-FFT) to convert the time-domain IF signal into the r-D space. A neighboring-reflection-points detection method is then developed to identify the region of interest (ROI) in the r-D space. Upon the identification of the ROI, the information of radial velocity and angle of the target is then extracted from both the radars. The extracted information is then used to synthesize the motion velocity feature, which is fed into an LSTM network for gesture classification.

### 4.1. r-D Region of Interest Detection

As the clutter of environmental background can significantly influence the results of target detection, the proposed algorithm first removes the cluttering signals and finds the ROI from the obtained movement data set to mitigate the impact of environmental background on gesture recognition.

With respect to the static part of the clutter, it is easy to eliminate its impact by averaging the different chirps of IF signals. Specifically, the reference signal, $y_{\mathrm{avg}}\in C^{N\times 1}$, can be obtained by summing and averaging the column vectors of the IF data matrix Y (see Section 3.1 for its definition): (14)$$\begin{array}{c} \begin{array}{c} y_{\mathrm{avg}}=\frac{1}{L}\sum_{l=1}^{L}Y\left[:,l\right]. \end{array} \end{array}$$

Then, after removing signals which are contributed by the static part of the clutter, $y_{\mathrm{avg}}$, from each column of the matrix Y, the remaining signals of the IF matrix, $Y_{\mathrm{remo}}\in C^{N\times L}$, containing the dynamic part of the clutter, e.g., reflection points of the moving target, can be expressed as: (15)$$\begin{array}{c} \begin{array}{c} Y_{\mathrm{remo}}\left[:,l\right]=Y\left[:,l\right]-y_{\mathrm{avg}},\;\;l\in \left\{1,\cdots ,L\right\}. \end{array} \end{array}$$

Compared to the elimination of the static part of the clutter, the signal-processing procedure for the dynamic clutter is more complicated. To reduce the impact of dynamic clutter on target detection, we develop a new detection method, as detailed in Algorithm 1. Figure 4 presents a schematic diagram for the signal-processing procedure of removing dynamic clutter. First of all, all the reflection points detected via the constant false alarm rate (CFAR) method are considered as potential target reflection points (red circles). Among potential target reflection points, the one with the highest velocity and being closest to the radar is chosen as the anchor point (red solid circle). Algorithm 1 is applied to search all potential reflection points (see the yellow circle) around the anchor point to form an ROI, and other potential target reflection points are removed in the following processes to reduce the impact of dynamic clutter.
**Algorithm 1** The neighboring-reflection-points detection for reducing dynamic clutter.**Input:** 
$Y_{\mathrm{remo}}$: the IF matrix after removing the static clutters**Output:**
the index of reflection point Q1:Applying the 2D-FFT introduced in Section 3.2 on $Y_{\mathrm{remo}}$ to obtain the matrix $\hat{Y}$2:Utilizing CFAR to calculate the dynamic threshold, ς. Then, save the column vector [i;j] into matrix D for $\hat{Y}\left(i,j\right)⩾\varsigma$3:$i_{\mathrm{central}}=\min\left(D\left(1,:\right)\right)$4:**for** 
k=1:L
**do**5:    **if** $\hat{Y}\left(i_{\mathrm{central}},k\right)==\max\left(\hat{Y}\left(i_{\mathrm{central}},:\right)\right)$ **then**6:        $j_{\mathrm{central}}=k$7:    **end if**8:**end for**9:$Q=[i_{\mathrm{central}};j_{\mathrm{central}}]$10:E=[1,0,−1,0;0,1,0,−1]11:${\mathrm{num}}_{1}=0;{\mathrm{num}}_{2}=1$12:**repeat**13:    **for** $l={\mathrm{num}}_{1}+1:{\mathrm{num}}_{2}$ **do**14:        **for** q=1:4 **do**15:           **if** Q(:,l)+expand(:,q)∈D&&Q(:,l)+E(:,q)∉Q **then**16:               Save [Q(:,l)+E(:,q)] into Q17:           **end if**18:        **end for**19:    **end for**20:    ${\mathrm{num}}_{1}={\mathrm{num}}_{2};{\mathrm{num}}_{2}=size\left(Q,2\right)$21:**until** 
${\mathrm{num}}_{1}=={\mathrm{num}}_{2}$

### 4.2. Feature Extraction

Upon the identification of the reflection points of the motion, the relative velocities and distances of the target, $v_{1}$, $d_{1}$, $v_{2}$, and $d_{2}$, can be acquired for the two radars using the traditional FMCW processing methods described in Section 3.2. Additionally, since both radars have multiple antennas, the angle of each reflection point, i.e., $\theta_{1}$ and $\theta_{2}$ with respect to the two radars, can also be obtained using beamforming techniques, specifically [35]: (16)$$\begin{array}{c} \begin{array}{c} y_{\mathrm{BF}}=\bar{y}\Psi , \end{array} \end{array}$$
where $\bar{y}\in C^{1\times M}$ denotes the IF data received at the antenna array and is presented as
(17)$$\begin{array}{c} \begin{array}{c} \bar{y}=\left[Y_{1}\left(n,l\right),Y_{2}\left(n,l\right),\cdots ,Y_{M}\left(n,l\right)\right], \end{array} \end{array}$$
and $Y_{M}\left(n,l\right)$ denotes the *n*-th row and *l*-th column IF data for *M*-th receiving antennas. In addition, the beam-steering vector with a certain determined angle is used to determine the angle of arrival of the target, and Ψ represents the beamforming steering matrix, which is given by: (18)$$\begin{array}{c} \begin{array}{c} \Psi =\left[\begin{array}{cccc} \psi_{1} & \psi_{2} & \cdots & \psi_{W} \end{array}\right], \end{array} \end{array}$$
where the beamformer used for the uniform linear array, $\psi_{w}$, can be represented as [35]: (19)$$\begin{array}{c} \begin{array}{c} \psi_{w}=\left[\begin{array}{c} 1\\ e^{-j\pi \cdot \cos[\Delta \theta \cdot (w-1)]}\\ \vdots \\ e^{-j\pi \cdot (M-1)\cdot \cos[\Delta \theta \cdot (w-1)]} \end{array}\right]. \end{array} \end{array}$$

In Equation (Equation 19), Δθ is the designed angular resolution of receiving beamforming and W=π/Δθ+1. Then, the angle of interest can be obtained by finding the peak value in $y_{\mathrm{BF}}$; i.e., supposing $W_{\mathrm{peak}}$ is the angle index of peak value, then the angle of target can be estimated as $\theta =(W_{\mathrm{peak}}-1)\times \Delta \theta -\pi /2$.

### 4.3. Motion Velocity Vector Synthesis

With the features extracted from these two radars, i.e., $\left(v_{1},\theta_{1}\right)$ and $\left(v_{2},\theta_{2}\right)$, the full velocity of the target can be estimated. Specifically, we consider a Cartesian coordinate system with the horizontal antenna direction of radar 1 as the *x*-axis. Then, the radial velocity vectors of the target detected at two radars 1 and 2 can be expressed as: (20)$$\begin{array}{c} \begin{array}{c} \vec{v_{n1}}=\vec{PF}=\left(v_{n1}\sin\theta_{1},v_{n1}\cos\theta_{1}\right), \end{array} \end{array}$$
and
(21)$$\begin{array}{c} \begin{array}{c} \vec{v_{n2}}=\vec{PE}=\left(v_{n2}\sin\theta_{2}^{\prime },v_{n2}\cos\theta_{2}^{\prime }\right), \end{array} \end{array}$$
respectively.

It can be noted that the normal directions of the antenna arrays of radar 1 and 2 do not need to be parallel to each other. As illustrated in Figure 5, radar 2 deployed at point B is equivalent to radar $2^{\prime }$ deployed at point C for estimating the same full velocity. Therefore, radar 2 can be equivalently considered as radar $2^{\prime }$, which is deployed in parallel to radar 1, and the relationship between these azimuths are given as follows: (22)$$\begin{array}{c} \begin{array}{c} \theta_{2}^{\prime }=\theta_{2}+\gamma . \end{array} \end{array}$$

Based on principles presented in Section 3.2, it can be seen that $v_{n}1$ and $v_{n}2$ are two projection components of *v* in two different directions. To obtain the true velocity vector of the target, we utilize the radial velocities, $v_{n}1$ and $v_{n}2$, as well as the azimuth angles $\theta_{1}$ and $\theta_{2}$, detected at different radars to resolve the true velocities of the target projected onto the *x*-axis and the *y*-axis, which are given by: (23)$$\begin{array}{c} \begin{array}{c} \left\{\begin{array}{cc} \; & y-v_{n}1\cos\theta_{1}=-\tan\theta_{1}\left(x-v_{n}1\sin\theta_{1}\right),\\ \; & y-v_{n}2\cos\theta_{2}^{\prime }=-\tan\theta_{2}\left(x-v_{n}2\sin\theta_{2}^{\prime }\right), \end{array}\right. \end{array} \end{array}$$
where the true velocity projects onto the *x*-axis, *x*, and can be expressed as: (24)$$\begin{array}{c} \begin{array}{cc} x\;=\; & v_{n1}\tan\theta_{1}\sin\theta_{1}+v_{n1}\cos\theta_{1}/\tan\theta_{1}-\tan\theta_{2}^{\prime }\\ & +v_{n2}\tan\theta_{2}^{\prime }\sin\theta_{2}^{\prime }+v_{n2}\cos\theta_{2}^{\prime }/\tan\theta_{2}^{\prime }-\tan\theta_{1}. \end{array} \end{array}$$

Finally, by substituting Equation (Equation 24) into Equation (Equation 23), we can obtain the value *y*, which represents the projection of the true velocity onto the *y*-axis.

The method described above is applied to each frame of the IF data to obtain the velocity of the target in the corresponding frame. After applying this process to κ consecutive frames, a sequence of the velocities, i.e., the motion velocity vector, is extracted and can be presented as: (25)$$\begin{array}{c} \begin{array}{c} V={\left[\begin{array}{cccc} x_{1,1} & x_{1,2} & \cdots & x_{1,\kappa }\\ x_{2,1} & x_{2,2} & \cdots & x_{2,\kappa } \end{array}\right]}^{T}. \end{array} \end{array}$$

### 4.4. Gesture Classification via LSTM

To classify the gestures based on the motion velocity vector, we adopt the LSTM network, which is widely used in sequential data analysis. As the motion velocity vector contains a sequence of velocities that capture the sequential movement of hand gestures, the LSTM is a natural fit to this task. The LSTM considered in this work contains two LSTM layers and one flatten layer, as illustrated in Figure 6.

Instead of making any modifications to the traditional structure of the LSTM network, we directly employ a general LSTM network which only has two LSTM layers. The input features V are passed through the flattening layer to reduce their dimensionality and subsequently subjected to the ReLU activation function.

## 5. Experiments and Results

In this section, we evaluate the performance of the dual-radar system driven by the information processing procedure detail in Section 4, using real data measured by radars.

### 5.1. Radar Parameters and Placement

In our experiments, we utilize two TI commercial mmWave FMCW radars for gesture recognition, e.g., one TI MMWCAS-DSP/RF-EVM radar and one TI IWR-6843-AOP-EVM radar.

The TI IWR-6843-AOP-EVM radar consists of three transmit antennas and four receive antennas and operates between 60 GHz and 64 GHz. The TI MMWCAS-DSP/RF-EVM consists of 12 transmit antennas and 16 receive antennas and operates between 76 GHz and 81 GHz. The deployment of the two mmWave FMCW radars and the environment for collecting the gesture data set are shown in Figure 7. The distance between these two radars is approximately 45 cm, as shown in Figure 7b, and the distance between radars and the hand of the gesture performers is approximately 80 cm, as shown in Figure 7c. With the aim of mitigating the influence of non-target objects on the recognition results, we ensure that the experiment encompassed predominantly wall structures within the background setting. The configuration parameters of radars used in our experiments are presented in Table 2 and Table 3.

### 5.2. Data Set

To evaluate the capability of the proposed framework in gesture recognition, we consider eight different gestures, as shown in Figure 8. The gestures can be classified into three categories, namely the *linear* gestures, the *curved* gestures and the *compound* ones. The linear gestures consist of four different basic gestures, e.g., left to right motion, right to left motion, front to back motion, and back to front motion, and the curved gestures contain those drawing a circle and drawing a semicircle. The compound gestures can be decomposed into a combination of several basic linear and curved gestures. For example, the compound gesture of drawing a “Z” in the plane consists of three basic gestures: left to right “→”, front to back “↙” and left to right “→”.

Based on the gestures defined, a data set with 1600 samples is collected in real experiments. To enhance the diversity of the gesture data set and to better test the robustness of the proposed framework, 10 volunteers (six males and four females) with different heights, weights, and gesture habits were invited to perform the gestures. For each volunteer, each gesture was performed 20 times.

The 1600 samples in the data set are divided randomly into two groups, with the training set containing 1280 samples and the test set containing 320 samples. The training set is used to training the LSTM network, while the test set is used to assess the accuracy of gesture recognition.

### 5.3. Performance Evaluations

In this subsection, we evaluate the performance of our proposed hand gesture recognition framework.

For comparisons, we consider two methods as baselines. Specifically, the first baseline adopts the conventional classification method based on the m-D image extracted from the radars and a pre-trained CNN. The m-D image is the input feature and the CNN network acts as a classifier. The second baseline converts the motion velocity vector to an image and then uses a pre-trained CNN network as a classifier.

For the first baseline, we consider three types of input feature for a more comprehensive evaluation, including the m-D images obtained from IWR-6843-AOP-EVM (shown in Figure 9a–h), the m-D image obtained from MMWCAS-DSP/RF-EVM (shown in Figure 9i–p) and the m-D images from both radars. For the first two types of input features, a classical CNN architecture with two convolutional layers is adopted. Each layer has 16 convolutional filters and the dimensions of each filter are 3 by 3. For the training of the considered CNN network, the number of epochs is set to 100 and the learning rate is considered as a constant value of 0.0005.

While both the m-D images are adopted as the input features, we train a two-channel CNN network to classify the gestures. For each channel of the CNN network, there are two convolutional layers, each with 32 filters and a kernel size of 5×5. The outputs of these two channels of the CNN network are concatenated and flattened, which are then fed into a fully connected layer to produce the gesture label. When training the network, the number of epochs and the learning rate are set to 50 and 0.0002, respectively.

For the second baseline, the image input to the CNN is an image converted from the motion velocity vector extracted from the two radars. To convert the images, we regard each column as a vector in Cartesian coordinates and plot them in chronological order. In Figure 10, we present several examples of the converted images. As can be seen from the figure, some gestures cannot be distinguished from the converted images, e.g., the right to left gesture and left to right gesture cannot be accurately distinguished from the generated images since they are almost same. The parameters of the CNN are identical to those used in baseline 1 with a single image as the input, but the CNN was trained using the motion images.

For the proposed solution, the size of the input feature is fixed at V into 20×2 with κ=20. The LSTM model has two LSTM layers, and each LSTM layer has 200 hidden units. ReLU is chosen as the activation function. In training process, the batch size is set to 3, while the number of epochs and learning rate are set to 100 and 0.005, respectively.

Table 4, Table 5, Table 6 and Table 7 present the confusion matrix to illustrate the recognition accuracy of the proposed framework and the baselines. As can be seen from comparing Table 4 to Table 5, the gesture recognition accuracy for the MMWCAS-DSP/RF-EVM radar is slightly higher compared to that of the IWR-6843-AOP-EVM radar, as expected, since the former has better signal quality. This is evident in Figure 9a–p, which show that MMWCAS-DSP/RF-EVM has lower background noise. However, for both radars, the conventional method using r-D images and CNN cannot offer satisfactory gesture recognition accuracy (from Table 4 and Table 5, the accuracies are 66.3% and 71.3%, respectively). Using the m-D images from both radars increases the accuracy to 75.3%. However, this performance is still inadequate for practical applications.

As a further enhancement, the second baseline, which utilizes the motion images and the CNN, achieves a significantly higher accuracy than those of baseline 1. As can be seen from Table 7, the gesture recognition accuracy is about 90%. These results show the effectiveness of the motion features extracted from the dual-radar system.

The results of our proposed framework are illustrated in Table 8. As can be seen from the table, our proposed framework achieves an accuracy of about 98%, which is significantly higher than the accuracy of all the baselines.

To further evaluate the robustness of our proposed gesture recognition framework against diverse gesture patterns, we perform the following experiment. In the experiment, the training set is built by using data sets extracted from eight randomly chosen volunteers, while the test set comes from the other two volunteers. In such circumstances, the LSTM network is trained only with a part of the volunteer data sets. The experimental results presented in Table 9 show a similar overall accuracy to the ones presented in Table 8, e.g., 97.5%.

To test the proposed framework against environmental diverseness, we introduce changes in the background environment for gesture recognition, which are demonstrated via a comparison between Figure 7a and Figure 11. In order to keep all other testing conditions unchanged compared to the previous experiment, we ensure that the distance between the two radars remains at 45 cm and the distance between radars and the hand remains at 80 cm.

To simulate a practical environment, several chairs are placed close to the performers of the gestures, as shown in Figure 11. Due to the implementation of the time division multiplexing mode in radar transmission during the experiment, the resulting beam coverage encompassed all regions within the radars’ FoVs. Consequently, non-target objects such as chairs exerted some influence on the recognition results.

By employing the same trained hyper-parameters of the LSTM network, as the previously conducted experiment, the results of which are presented in Table 8, 160 samples for eight gestures are inputted for recognition, and the result for the background-changed comparison is 98.1%, as illustrated in Table 10, which mirrors the gesture recognition performance given in Table 8.

These comparison results validate the robustness of our proposed gesture recognition algorithm and demonstrate its resilience towards the changes in the background environment.

## 6. Discussion and Conclusions

We primarily discussed two potential sources of error in the experiments. Firstly, we control the number of LSTM layers to satisfy the requirement of low complexity, which will influence the accuracy of recognition. The accuracy can be further improved by increasing the number of LSTM layers or adopting other neural networks with higher complexity. Another factor contributing to errors is the resemblance between the gestures due to arbitrary drawing. For example, when volunteers execute the second stage of the gesture of drawing an “L”, in which the hand moves from left to right, arbitrary drawing leads to a short distance of motion for this specific stage. Consequently, this marginal tangential displacement fails to exhibit pronounced Doppler frequency, thereby leading to a final result that resembles the gesture of moving from front to back.

Furthermore, there are three potential research directions that can be explored in the future development of this system. Regarding the application scenarios of the system, we hope to apply this system in scenarios with multiple people and hands while further improving the recognition accuracy, thus placing higher demands on the extraction of hand reflection points. Significantly, when dealing with the recognition of two hand gestures, a pressing challenge arises in effectively tracking the reflective points of the hands and synthesizing motion velocity vectors when confronted with overlapping trajectories.

In terms of recognizing the types of gestures by the system, our system is currently designed for macro-gestures, and it is hoped that the ability to recognize micro-gestures can be developed by maintaining the utilization of our motion velocity vector algorithm in the recognition process. Given the inherently small magnitude of displacements exhibited by micro-gestures, the extraction of precise velocity and azimuth information is a notable challenge.

For the system hardware, when more than two radars are present in the same space, the synthesis algorithm for radar data necessitates addressing several challenges, including, but not limited to, minimizing placement restrictions of the radars, synchronizing and transmitting radar data, and selecting the most appropriate radar data to be utilized.

In this work, we have proposed a novel gesture recognition framework based on two mmWave FMCW radars. A novel motion velocity vector synthesis algorithm has been developed to extract the gesture characteristics that are crucial for gesture recognition. The proposed motion velocity synthesis algorithm allows for flexible deployment of the mmWave radars and does not require knowledge of the relative position of the two radars. Based on the motion velocity vector, a gesture recognition model based on an LSTM network has been designed. The numerical results from real measurements demonstrate that the proposed framework can achieve satisfactory hand gesture recognition accuracy, even when there is a change in the environment and the training data and the test data are collected from different volunteers.

## Figures and Tables

**Figure 1 sensors-23-08551-f001:**
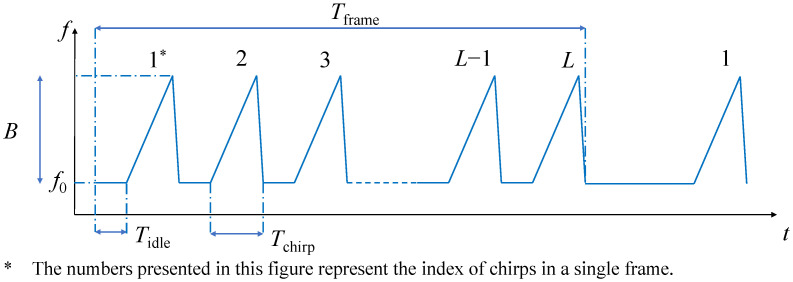
The time frame structure of the transmitted FMCW signal.

**Figure 2 sensors-23-08551-f002:**
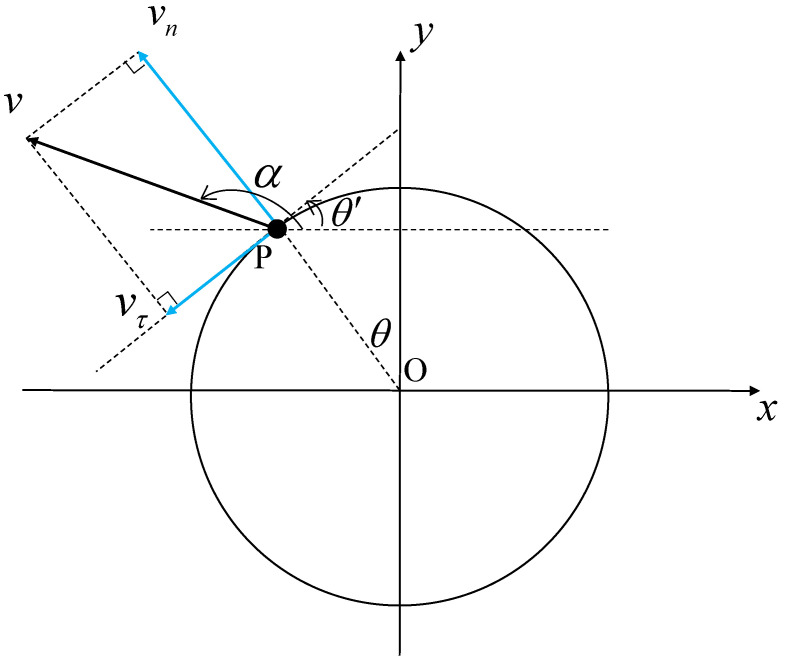
The decomposition relationship between the absolute velocity and the radial velocity for gestures.

**Figure 3 sensors-23-08551-f003:**
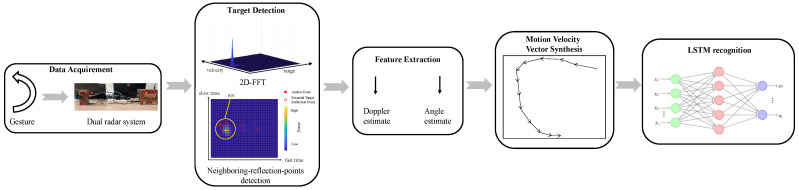
The flow chart of the proposed gesture recognition system.

**Figure 4 sensors-23-08551-f004:**
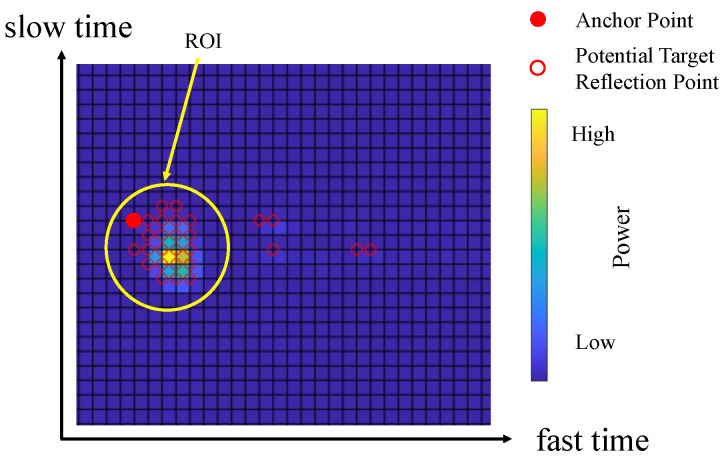
The demonstration for the proposed neighboring-reflection-points detection method.

**Figure 5 sensors-23-08551-f005:**
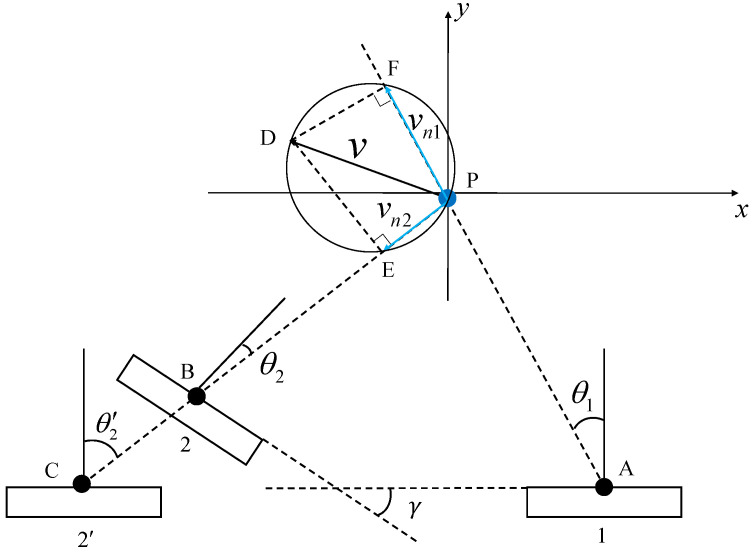
Motion velocity vector synthesis.

**Figure 6 sensors-23-08551-f006:**
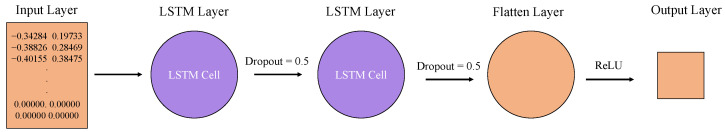
The architecture of the LSTM network adopted in the proposed gesture recognition system.

**Figure 7 sensors-23-08551-f007:**
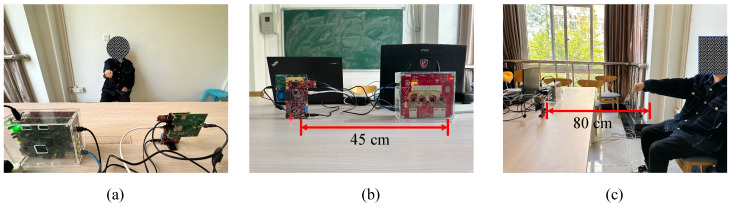
The environment of the experiment and the placement of two radars: (**a**) experimental background; (**b**) specific placement of radars; (**c**) user-to-radar spacing.

**Figure 8 sensors-23-08551-f008:**
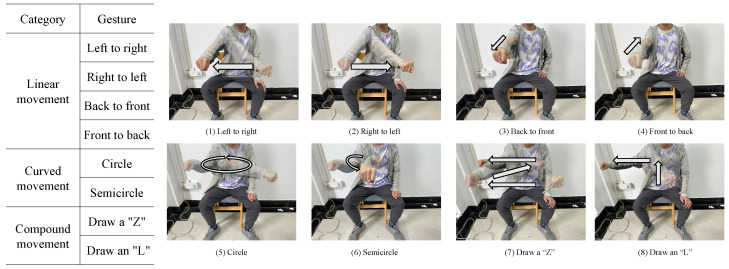
Diagram of the eight hand gestures.

**Figure 9 sensors-23-08551-f009:**
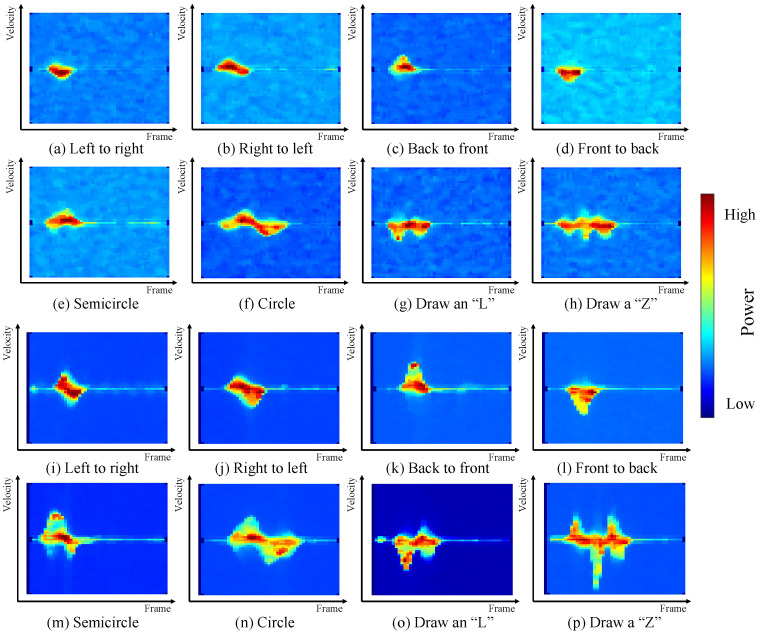
The m-D images of different hand gestures: (**a**–**h**) obtained from IWR-6843-AOP-EVM; (**i**–**p**) obtained from MMWCAS-RF-EVM.

**Figure 10 sensors-23-08551-f010:**
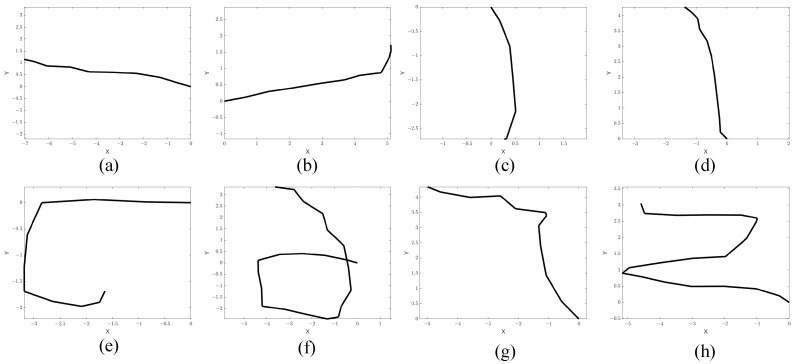
Grayscale images of eight different trajectories: (**a**) left to right, (**b**) right to left, (**c**) back to front, (**d**) front to back, (**e**) semicircle, (**f**) circle, (**g**) draw an “L”, (**h**) draw a “Z”. “X” and “Y” in figures represent the value of the motion velocity vector in the x and y directions, respectively.

**Figure 11 sensors-23-08551-f011:**
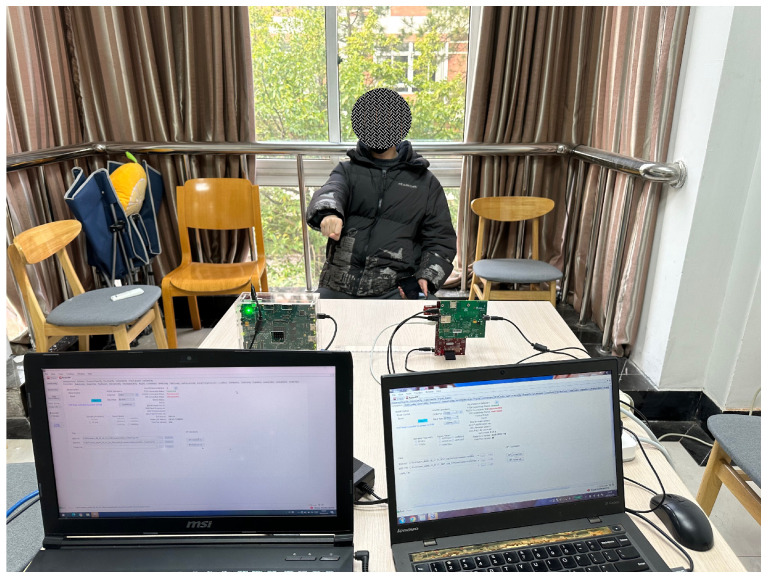
Test environment and radar placement.

**Table 1 sensors-23-08551-t001:** A list of notations.

Parameter	Symbol	Calculation Formula	Parameter	Symbol
Sampling Frequency	$f_{s}$	$f_{s}=N/T_{\mathrm{chirp}}$	Bandwidth	*B*
Sampling Interval	$T_{s}$	$T_{s}=1/f_{s}$	Samples per Chirp	*N*
Modulation Time per Chirp	$T_{\mathrm{chirp}}$	$T_{\mathrm{chirp}}=N/f_{s}$	Chirps per Frame	*L*
Frequency Modulation Slope	*K*	$K=B/T_{\mathrm{chirp}}$	Idle Time	$T_{\mathrm{idle}}$
Time per Frame	$T_{\mathrm{frame}}$	$T_{\mathrm{frame}}=N\times \left(T_{\mathrm{idle}}+T_{\mathrm{chirp}}\right)$	Start Frequency	$f_{0}$

**Table 2 sensors-23-08551-t002:** Radar configuration parameters for IWR-6843-AOP-EVM.

Parameter	Symbol	Value	Parameter	Symbol	Value
Bandwidth	*B*	3999.35 MHz	Sampling Frequency	$f_{s}$	8000 ksps
Sampling Interval	$T_{s}$	$1.25\times 10^{-7}$ s	Samples per Chirp	*N*	128
Modulation Time per Chirp	$T_{\mathrm{chirp}}$	50 μs	Chirps per Frame	*L*	64
Time per Frame	$T_{\mathrm{frame}}$	100 ms	Number of Frames	*P*	50
Frequency Modulation Slope	*K*	79.987 MHz/μs	Idle Time	$T_{\mathrm{idle}}$	300 μs
Start Frequency	$f_{0}$	60 GHz	Time per chirp	$T=T_{\mathrm{chirp}}+T_{\mathrm{idle}}$	350 μs

**Table 3 sensors-23-08551-t003:** Radar configuration parameters for MMWCAS-DSP/RF-EVM.

Parameter	Symbol	Value	Parameter	Symbol	Value
Bandwidth	*B*	3200 MHz	Sampling Frequency	$f_{s}$	8000 ksps
Sampling Interval	$T_{s}$	$1.25\times 10^{-7}$ s	Samples per Chirp	*N*	128
Modulation Time per Chirp	$T_{\mathrm{chirp}}$	40 μs	Chirps per Frame	*L*	64
Time per Frame	$T_{\mathrm{frame}}$	100 ms	Number of Frames	*P*	50
Frequency Modulation Slope	*K*	80 MHz/μs	Idle Time	$T_{\mathrm{idle}}$	5 μs
Start Frequency	$f_{0}$	77 GHz	Time per Chirp	$T=T_{\mathrm{chirp}}+T_{\mathrm{idle}}$	45 μs

**Table 4 sensors-23-08551-t004:** Confusion matrix for gesture recognition with CNN and m-D images from IWR-6843-AOP-EVM.

		Prediction								
		Left to Right	Right to Left	Back to Front	Front to Back	Semicircle	Circle	Draw an “L”	Draw a “Z”	Accuracy
Ground Truth	Left to right	26	4	4	3	1	2	3	0	60.5%
	Right to left	2	30	3	0	1	2	0	4	71.4%
	Back to front	2	3	27	2	0	2	1	2	69.2%
	Front to back	1	1	3	25	2	3	2	3	62.5%
	Semicircle	2	2	2	1	24	1	4	0	66.7%
	Circle	0	4	3	2	3	24	1	2	61.5%
	Draw an “L”	0	2	1	1	1	2	25	4	69.4%
	Draw a “Z”	1	2	2	2	3	1	3	31	68.9%
	Accuracy	76.5%	62.5%	60.0%	69.4%	68.6%	64.9%	64.1%	67.4%	66.3%

**Table 5 sensors-23-08551-t005:** Confusion matrix for gesture recognition with CNN and m-D images from MMWCAS-DSP/RF-EVM.

		Prediction								
		Left to Right	Right to Left	Back to Front	Front to Back	Semicircle	Circle	Draw an “L”	Draw a “Z”	Accuracy
Ground Truth	Left to right	21	0	4	4	2	2	1	1	60.0%
	Right to left	2	28	2	1	0	4	0	3	70.0%
	Back to front	3	1	32	1	0	0	0	0	86.5%
	Front to back	4	2	4	35	3	2	0	1	68.6%
	Semicircle	1	1	4	4	23	0	2	2	62.2%
	Circle	1	0	0	1	1	33	2	1	84.6%
	Draw an “L”	4	2	1	0	2	2	25	5	61.0%
	Draw a “Z”	0	2	1	0	3	1	2	31	77.5%
	Accuracy	58.3%	77.8%	66.7%	76.1%	67.6%	75.0%	78.1%	70.5%	71.3%

**Table 6 sensors-23-08551-t006:** Confusion matrix for gesture recognition with CNN and joint utilization of m-D images from two radars.

		Prediction								
		Left to Right	Right to Left	Back to Front	Front to Back	Semicircle	Circle	Draw an “L”	Draw a “Z”	Accuracy
Ground Truth	Left to right	24	2	4	3	1	1	2	3	60.0%
	Right to left	2	33	0	1	1	2	0	2	80.5%
	Back to front	0	1	32	0	1	0	0	1	91.4%
	Front to back	1	0	3	30	2	1	1	2	75.0%
	Semicircle	3	0	2	3	30	3	1	3	66.7%
	Circle	0	2	2	0	3	28	0	1	77.8%
	Draw an “L”	1	1	1	2	0	0	30	3	78.9%
	Draw a “Z”	1	0	1	2	5	0	2	34	75.6%
	Accuracy	75.0%	84.6%	71.1%	73.2%	69.8%	80.0%	83.3%	69.4%	75.3%

**Table 7 sensors-23-08551-t007:** Confusion matrix for the recognition with motion images.

		Prediction						
		Left and Right	Back and Front	Semicircle	Circle	Draw an “L”	Draw a “Z”	Accuracy
Ground Truth	Left and Right	80	1	0	0	0	0	98.8%
	Back and Front	0	70	1	0	0	0	98.6%
	Semicircle	3	0	42	0	0	1	91.3%
	Circle	0	9	4	29	0	1	67.4%
	Draw an “L”	0	0	0	0	28	0	100%
	Draw a “Z”	2	0	1	7	1	40	78.4%
	Accuracy	94.1%	87.5%	87.5%	80.6%	86.6%	95.2%	90.3%

**Table 8 sensors-23-08551-t008:** Confusion matrix for gesture recognition with synthesized motion vectors.

		Prediction								
		Left to Right	Right to Left	Back to Front	Front to Back	Semicircle	Circle	Draw an “L”	Draw a “Z”	Accuracy
Ground Truth	Left to right	26	0	1	0	0	0	0	0	96.3%
	Right to left	0	28	0	0	0	0	0	0	100.0%
	Back to front	0	0	35	0	0	0	0	0	100.0%
	Front to back	1	0	0	35	0	0	0	1	94.6%
	Semicircle	0	0	0	0	33	0	0	0	100.0%
	Circle	0	1	0	0	0	29	0	1	93.5%
	Draw an “L”	0	0	0	0	0	0	33	0	100.0%
	Draw a “Z”	0	0	0	0	0	0	0	32	100.0%
	Accuracy	96.3%	96.6%	97.2%	100.0%	100.0%	100.0%	100.0%	94.1%	98.0%

**Table 9 sensors-23-08551-t009:** Confusion matrix for gesture recognition with synthesized motion vectors (two users as testing set).

		Prediction								
		Left to Right	Right to Left	Back to Front	Front to Back	Semicircle	Circle	Draw an “L”	Draw a “Z”	Accuracy
Ground Truth	Left to right	40	0	0	0	0	0	0	0	100.0%
	Right to left	0	40	0	0	0	0	0	0	100.0%
	Back to front	0	0	40	0	0	0	0	0	100.0%
	Front to back	0	0	0	39	0	0	1	0	97.5%
	Semicircle	0	0	2	0	38	0	0	0	95.0%
	Circle	0	1	0	0	0	39	0	0	97.5%
	Draw an “L”	0	0	0	3	0	0	37	0	92.5%
	Draw a “Z”	1	0	0	0	0	0	0	39	97.5%
	Accuracy	97.6%	97.6%	95.2%	92.9%	100.0%	100.0%	97.4%	100.0%	97.5%

**Table 10 sensors-23-08551-t010:** Confusion matrix for the recognition with vector pairs (change in background environment).

		Prediction								
		Left to Right	Right to Left	Back to Front	Front to Back	Semicircle	Circle	Draw an “L”	Draw a “Z”	Accuracy
Ground Truth	Left to right	20	0	0	0	0	0	0	0	100.0%
	Right to left	0	20	0	0	0	0	0	0	100.0%
	Back to front	0	0	20	0	0	0	0	0	100.0%
	Front to back	0	0	0	20	0	0	0	0	100.0%
	Semicircle	0	0	0	0	20	0	0	0	100.0%
	Circle	0	0	0	0	2	18	0	0	90.0%
	Draw an “L”	0	0	0	1	0	0	19	0	95.0%
	Draw a “Z”	0	0	0	0	0	0	0	20	100.0%
	Accuracy	100.0%	100.0%	100.0%	95.2%	90.9%	100.0%	100.0%	100.0%	98.1%

## Data Availability

The data presented in this study are available upon request from the corresponding author.

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
