# Peer review of "A Low-Complexity Hand Gesture Recognition Framework via Dual mmWave FMCW Radar System"

_sensors, 2023, doi:10.3390/s23208551_

Round 1
Reviewer 1 Report
The topics discussed in the work are scientifically interesting and necessary. But it is generally known in literature.
The article presents the state of knowledge in the literature on the topic of the work.
The article lacks literature references to mathematical formulas, which makes it difficult for the reviewer to assess which mathematical formulas are original. I did not find any significant substantive shortcomings in the article.
Conlusions contain too little information and should be corrected.
Author Response
Please find the response letter attached in the following.

Reviewer 2 Report
The developed new pattern recognition system is close to practical implementation. This is extremely important for any instrument complexes. Despite a number of minor shortcomings, I recommend the article to print. The only thing I want to see in the article is ways to improve this system. I recommend the authors to expand the conclusion, in which it is necessary to indicate the directions of further development of the developed system.
Author Response

(The authors gave the same response as above.)

Reviewer 3 Report
The authors have applied dual FMCW radar for a gesture recognition, and obtained improved results compared to previous method. I think this manuscript worth publishing in Applied Sciences, the Special Issue "Advances in Doppler and FMCW Radar Sensors". There are a few comments as shown in the followings.
1. Lines 359-363: The authors state that for evaluation of the robustness the data set extracted from eight volunteers - - - . The results is shown in Table 10.
Please describe a little more comprehensive manner. Also please describe the conditions for the initial experiments, whose results are shown in Tables 4-9.
2. Lines 368-377: The authors try to perform an experiment simulating the practical environment. Since chairs are stationary objects, I think, they do not affect the results much. If we have to worry about multiple reflection, it depends on the directivity of the transmitting beam. Please describe the shape of transmitting beam.
3. Figures 9 and 10: Please describe the horizontal and vertical axes.
4. Figure 3: The text in figure is hard to read. Please make them bigger.
Author Response

(The authors gave the same response as above.)
